# Parallel Streaming Wasserstein Barycenters

**Matthew Staib**
MIT CSAIL
mstaib@mit.edu

**Sebastian Claici**
MIT CSAIL
sclaici@mit.edu

**Justin Solomon**
MIT CSAIL
jsolomon@mit.edu

**Stefanie Jegelka**
MIT CSAIL
stefje@mit.edu

## Abstract

Efficiently aggregating data from different sources is a challenging problem, particularly when samples from each source are distributed differently. These differences can be inherent to the inference task or present for other reasons: sensors in a sensor network may be placed far apart, affecting their individual measurements. Conversely, it is computationally advantageous to split Bayesian inference tasks across subsets of data, but data need not be identically distributed across subsets. One principled way to fuse probability distributions is via the lens of optimal transport: the Wasserstein barycenter is a single distribution that summarizes a collection of input measures while respecting their geometry. However, computing the barycenter scales poorly and requires discretization of all input distributions and the barycenter itself. Improving on this situation, we present a scalable, communication-efficient, parallel algorithm for computing the Wasserstein barycenter of arbitrary distributions. Our algorithm can operate directly on continuous input distributions and is optimized for streaming data. Our method is even robust to nonstationary input distributions and produces a barycenter estimate that tracks the input measures over time. The algorithm is semi-discrete, needing to discretize only the barycenter estimate. To the best of our knowledge, we also provide the first bounds on the quality of the approximate barycenter as the discretization becomes finer. Finally, we demonstrate the practical effectiveness of our method, both in tracking moving distributions on a sphere, as well as in a large-scale Bayesian inference task.

## 1 Introduction

A key challenge when scaling up data aggregation occurs when data comes from multiple sources, each with its own inherent structure. Sensors in a sensor network may be configured differently or placed far apart, but each individual sensor simply measures a different view of the same quantity. Similarly, user data collected by a server in California will differ from that collected by a server in Europe: the data samples may be independent but are not identically distributed.

One reasonable approach to aggregation in the presence of multiple data sources is to perform inference on each piece independently and fuse the results. This is possible when the data can be distributed randomly, using methods akin to distributed optimization [52, 53]. However, when the data is *not* split in an i.i.d. way, Bayesian inference on different subsets of observed data yields slightly different "subset posterior" distributions for each subset that must be combined [33]. Further complicating matters, data sources may be nonstationary. How can we fuse these different data sources for joint analysis in a consistent and structure-preserving manner?

We address this question using ideas from the theory of *optimal transport*. Optimal transport gives us a principled way to measure distances between measures that takes into account the underlying space on which the measures are defined. Intuitively, the optimal transport distance between two distributions measures the amount of *work* one would have to do to move all mass from one distribution to the other. Given $J$ input measures $\{\mu_j\}_{j=1}^J$, it is natural, in this setting, to ask for a measure $\nu$ that minimizes the total squared distance to the input measures. This measure $\nu$ is called the *Wasserstein*

*barycenter* of the input measures [1], and should be thought of as an aggregation of the input measures which preserves their geometry. This particular aggregation enjoys many nice properties: in the earlier Bayesian inference example, aggregating subset posterior distributions via their Wasserstein barycenter yields guarantees on the original inference task [47].

If the measures $\mu_j$ are discrete, their barycenter can be computed relatively efficiently via either a sparse linear program [2], or regularized projection-based methods [16, 7, 51, 17]. However, **1.** these techniques scale poorly with the support of the measures, and quickly become impractical as the support becomes large. **2.** When the input measures are continuous, to the best of our knowledge the only option is to discretize them via sampling, but the rate of convergence to the true (continuous) barycenter is not well-understood. These two confounding factors make it difficult to utilize barycenters in scenarios like parallel Bayesian inference where the measures are continuous and a fine approximation is needed. These are the primary issues we work to address in this paper.

Given sample access to $J$ potentially continuous distributions $\mu_j$, we propose a communication-efficient, parallel algorithm to estimate their barycenter. Our method can be parallelized to $J$ worker machines, and the messages sent between machines are merely single integers. We require a discrete approximation only of the barycenter itself, making our algorithm *semi-discrete*, and our algorithm scales well to fine approximations (e.g. $n \approx 10^6$). In contrast to previous work, we provide guarantees on the quality of the approximation as $n$ increases. These rates apply to the general setting in which the $\mu_j$'s are defined on manifolds, with applications to directional statistics [46]. Our algorithm is based on stochastic gradient descent as in [22] and hence is robust to gradual changes in the distributions: as the $\mu_j$'s change over time, we maintain a moving estimate of their barycenter, a task which is not possible using current methods without solving a large linear program in each iteration.

We emphasize that we aggregate the input distributions into a summary, the barycenter, which is itself a distribution. Instead of performing any single domain-specific task such as clustering or estimating an expectation, we can simply compute the barycenter of the inputs and process it later any arbitrary way. This generality coupled with the efficiency and parallelism of our algorithm yields immediate applications in fields from large scale Bayesian inference to e.g. streaming sensor fusion.

**Contributions.** **1.** We give a communication-efficient and fully parallel algorithm for computing the barycenter of a collection of distributions. Although our algorithm is semi-discrete, we stress that the input measures can be *continuous*, and even *nonstationary*. **2.** We give bounds on the quality of the recovered barycenter as our discretization becomes finer. These are the first such bounds we are aware of, and they apply to measures on arbitrary compact and connected manifolds. **3.** We demonstrate the practical effectiveness of our method, both in tracking moving distributions on a sphere, as well as in a real large-scale Bayesian inference task.

## 1.1 Related work

**Optimal transport.** A comprehensive treatment of optimal transport and its many applications is beyond the scope of our work. We refer the interested reader to the detailed monographs by Villani [49] and Santambrogio [42]. Fast algorithms for optimal transport have been developed in recent years via Sinkhorn's algorithm [15] and in particular stochastic gradient methods [22], on which we build in this work. These algorithms have enabled several applications of optimal transport and Wasserstein metrics to machine learning, for example in supervised learning [21], unsupervised learning [34, 5], and domain adaptation [14]. Wasserstein barycenters in particular have been applied to a wide variety of problems including fusion of subset posteriors [47], distribution clustering [51], shape and texture interpolation [45, 40], and multi-target tracking [6].

When the distributions $\mu_j$ are discrete, transport barycenters can be computed relatively efficiently via either a sparse linear program [2] or regularized projection-based methods [16, 7, 51, 17]. In settings like posterior inference, however, the distributions $\mu_j$ are likely continuous rather than discrete, and the most obvious viable approach requires discrete approximation of each $\mu_j$. The resulting discrete barycenter converges to the true, continuous barycenter as the approximations become finer [10, 28], but the rate of convergence is not well-understood, and finely approximating each $\mu_j$ yields a very large linear program.

**Scalable Bayesian inference.** Scaling Bayesian inference to large datasets has become an important topic in recent years. There are many approaches to this, ranging from parallel Gibbs sampling [38, 26]

to stochastic and streaming algorithms [50, 13, 25, 12]. For a more complete picture, we refer the reader to the survey by Angelino et al. [3].

One promising method is via subset posteriors: instead of sampling from the posterior distribution given by the full data, the data is split into smaller tractable subsets. Performing inference on each subset yields several subset posteriors, which are biased but can be combined via their Wasserstein barycenter [47], with provable guarantees on approximation quality. This is in contrast to other methods that rely on summary statistics to estimate the true posterior [33, 36] and that require additional assumptions. In fact, our algorithm works with arbitrary measures and on manifolds.

## 2   Background

Let $(\mathcal{X}, d)$ be a metric space. Given two probability measures $\mu \in \mathcal{P}(\mathcal{X})$ and $\nu \in \mathcal{P}(\mathcal{X})$ and a cost function $c : \mathcal{X} \times \mathcal{X} \to [0, \infty)$, the Kantorovich optimal transport problem asks for a solution to

$$\inf \left\{ \int_{\mathcal{X} \times \mathcal{X}} c(x, y) d\gamma(x, y) : \gamma \in \Pi(\mu, \nu) \right\} \tag{1}$$

where $\Pi(\mu, \nu)$ is the set of measures on the product space $\mathcal{X} \times \mathcal{X}$ whose marginals evaluate to $\mu$ and $\nu$, respectively.

Under mild conditions on the cost function (lower semi-continuity) and the underlying space (completeness and separability), problem (1) admits a solution [42]. Moreover, if the cost function is of the form $c(x, y) = d(x, y)^p$, the optimal transportation cost is a distance metric on the space of probability measures. This is known as the *Wasserstein distance* and is given by

$$W_p(\mu, \nu) = \left( \inf_{\gamma \in \Pi(\mu, \nu)} \int_{\mathcal{X} \times \mathcal{X}} d(x, y)^p d\gamma(x, y) \right)^{1/p}. \tag{2}$$

Optimal transport has recently attracted much attention in machine learning and adjacent communities [21, 34, 14, 39, 41, 5]. When $\mu$ and $\nu$ are discrete measures, problem (2) is a linear program, although faster regularized methods based on Sinkhorn iteration are used in practice [15]. Optimal transport can also be computed using stochastic first-order methods [22].

Now let $\mu_1, \ldots, \mu_J$ be measures on $\mathcal{X}$. The Wasserstein barycenter problem, introduced by Agueh and Carlier [1], is to find a measure $\nu \in \mathcal{P}(\mathcal{X})$ that minimizes the functional

$$F[\nu] := \frac{1}{J} \sum_{j=1}^{J} W_2^2(\mu_j, \nu). \tag{3}$$

Finding the *barycenter* $\nu$ is the primary problem we address in this paper. When each $\mu_j$ is a discrete measure, the exact barycenter can be found via linear programming [2], and many of the regularization techniques apply for approximating it [16, 17]. However, the problem size grows quickly with the size of the support. When the measures $\mu_j$ are truly continuous, we are aware of only one strategy: sample from each $\mu_j$ in order to approximate it by the empirical measure, and then solve the discrete barycenter problem.

We directly address the problem of computing the barycenter when the input measures can be continuous. We solve a semi-discrete problem, where the target measure is a finite set of points, but we do not discretize any other distribution.

## 3   Algorithm

We first provide some background on the dual formulation of optimal transport. Then we derive a useful form of the barycenter problem, provide an algorithm to solve it, and prove convergence guarantees. Finally, we demonstrate how our algorithm can easily be parallelized.

### 3.1   Mathematical preliminaries

The primal optimal transport problem (1) admits a dual problem [42]:

$$OT_c(\mu, \nu) = \sup_{v \text{ 1-Lipschitz}} \left\{ \mathbb{E}_{Y \sim \nu}[v(Y)] + \mathbb{E}_{X \sim \mu}[v^c(X)] \right\}, \tag{4}$$

where $v^c(x) = \inf_{y \in \mathcal{X}}\{c(x,y) - v(y)\}$ is the *c-transform* of $v$ [49]. When $\nu = \sum_{i=1}^{n} w_i \delta_{y_i}$ is discrete, problem (4) becomes the *semi-discrete* problem

$$OT_c(\mu, \nu) = \max_{v \in \mathbb{R}^n} \left\{ \langle w, v \rangle + \mathbb{E}_{X \sim \mu}[h(X,v)] \right\}, \tag{5}$$

where we define $h(x,v) = v^c(x) = \min_{i=1,\ldots,n}\{c(x,y_i) - v_i\}$. Semi-discrete optimal transport admits efficient algorithms [31, 29]; Genevay et al. [22] in particular observed that given sample oracle access to $\mu$, the semi-discrete problem can be solved via stochastic gradient ascent. Hence optimal transport distances can be estimated even in the semi-discrete setting.

## 3.2 Deriving the optimization problem

Absolutely continuous measures can be approximated arbitrarily well by discrete distributions with respect to Wasserstein distance [30]. Hence one natural approach to the barycenter problem (3) is to approximate the true barycenter via discrete approximation: we fix $n$ support points $\{y_i\}_{i=1}^n \in \mathcal{X}$ and search over assignments of the mass $w_i$ on each point $y_i$. In this way we wish to find the discrete distribution $\nu_n = \sum_{i=1}^n w_i \delta_{y_i}$ with support on those $n$ points which optimizes

$$\min_{w \in \Delta_n} F(w) = \min_{w \in \Delta_n} \frac{1}{J} \sum_{j=1}^{J} W_2^2(\mu_j, \nu_n) \tag{6}$$

$$= \min_{w \in \Delta_n} \left\{ \frac{1}{J} \sum_{j=1}^{J} \max_{v^j \in \mathbb{R}^n} \left\{ \langle w, v^j \rangle + \mathbb{E}_{X_j \sim \mu_j}[h(X_j, v^j)] \right\} \right\}. \tag{7}$$

where we have defined $F(w) := F[\nu_n] = F[\sum_{i=1}^n w_i \delta_{y_i}]$ and used the dual formulation from equation (5). in Section 4, we discuss the effect of different choices for the support points $\{y_i\}_{i=1}^n$.

Noting that the variables $v^j$ are uncoupled, we can rearrange to get the following problem:

$$\min_{w \in \Delta_n} \max_{v^1,\ldots,v^J} \frac{1}{J} \sum_{j=1}^{J} \left[ \langle w, v^j \rangle + \mathbb{E}_{X_j \sim \mu_j}[h(X_j, v^j)] \right]. \tag{8}$$

Problem (8) is convex in $w$ and jointly concave in the $v^j$, and we can compute an unbiased gradient estimate for each by sampling $X_j \sim \mu_j$. Hence, we could solve this saddle-point problem via simultaneous (sub)gradient steps as in Nemirovski and Rubinstein [37]. Such methods are simple to implement, but in the current form we must project onto the simplex $\Delta_n$ at each iteration. This requires only $O(n \log n)$ time [24, 32, 19] but makes it hard to decouple the problem across each distribution $\mu_j$. Fortunately, we can reformulate the problem in a way that avoids projection entirely. By strong duality, Problem (8) can be written as

$$\max_{v^1,\ldots,v^J} \min_{w \in \Delta_n} \left\{ \left\langle \frac{1}{J} \sum_{j=1}^{J} v^j, w \right\rangle + \frac{1}{J} \sum_{j=1}^{J} \mathbb{E}_{X_j \sim \mu_j}[h(X_j, v^j)] \right\} \tag{9}$$

$$= \max_{v^1,\ldots,v^J} \left\{ \min_i \left\{ \frac{1}{J} \sum_{j=1}^{J} v_i^j \right\} + \frac{1}{J} \sum_{j=1}^{J} \mathbb{E}_{X_j \sim \mu_j}[h(X_j, v^j)] \right\}. \tag{10}$$

Note how the variable $w$ disappears: for any fixed vector $b$, minimization of $\langle b, w \rangle$ over $w \in \Delta_n$ is equivalent to finding the minimum element of $b$. The optimal $w$ can also be computed in closed form when the barycentric cost is entropically regularized as in [9], which may yield better convergence rates but requires dense updates that, e.g., need more communication in the parallel setting. In either case, we are left with a concave maximization problem in $v^1, \ldots, v^J$, to which we can directly apply stochastic gradient ascent. Unfortunately the gradients are still not sparse and decoupled. We obtain sparsity after one final transformation of the problem: by replacing each $\sum_{j=1}^{J} v_i^j$ with a variable $s_i$ and enforcing this equality with a constraint, we turn problem (10) into the constrained problem

$$\max_{s,v^1,\ldots,v^J} \frac{1}{J} \sum_{j=1}^{J} \left[ \frac{1}{J} \min_i s_i + \mathbb{E}_{X_j \sim \mu_j}[h(X_j, v^j)] \right] \quad \text{s.t.} \quad s = \sum_{j=1}^{J} v^j. \tag{11}$$

## 3.3 Algorithm and convergence

We can now solve this problem via stochastic projected subgradient ascent. This is described in Algorithm 1; note that the sparse adjustments after the gradient step are actually projections onto the constraint set with respect to the $\ell_1$ norm. Derivation of this sparse projection step is given rigorously in Appendix A. Not only do we have an optimization algorithm with sparse updates, but we can even recover the optimal weights $w$ from standard results in online learning [20]. Specifically, in a zero-sum game where one player plays a no-regret learning algorithm and the other plays a best-response strategy, the average strategies of both players converge to optimal:

**Theorem 3.1.** *Perform $T$ iterations of stochastic subgradient ascent on $u = (s, v^1, \ldots, v^J)$ as in Algorithm 1, and use step size $\gamma = \frac{R}{4\sqrt{T}}$, assuming $\|u_t - u^*\|_1 \leq R$ for all $t$. Let $i_t$ be the minimizing index chosen at iteration $t$, and write $\overline{w}_T = \frac{1}{T} \sum_{t=1}^T e_{i_t}$. Then we can bound*

$$\mathbb{E}[F(\overline{w}_T) - F(w^*)] \leq 4R/\sqrt{T}. \tag{12}$$

*The expectation is with respect to the randomness in the subgradient estimates $g_t$.*

Theorem 3.1 is proved in Appendix B. The proof combines the zero-sum game idea above, which itself comes from [20], with a regret bound for online gradient descent [54, 23].

---

**Algorithm 1** Subgradient Ascent

$s, v^1, \ldots, v^J \leftarrow 0_n$
**loop**
    Draw $j \sim \text{Unif}[1, \ldots, J]$
    Draw $x \sim \mu_j$
    $i_W \leftarrow \text{argmin}_i \{c(x, y_i) - v_i^j\}$
    $i_M \leftarrow \text{argmin}_i s_i$
    $v_{i_W}^j \leftarrow v_{i_W}^j - \gamma$     ▷ Gradient update
    $s_{i_M} \leftarrow s_{i_M} + \gamma/J$   ▷ Gradient update
    $v_{i_W}^j \leftarrow v_{i_W}^j + \gamma/2$         ▷ Projection
    $v_{i_M}^j \leftarrow v_{i_M}^j + \gamma/(2J)$     ▷ Projection
    $s_{i_W} \leftarrow s_{i_W} - \gamma/2$        ▷ Projection
    $s_{i_M} \leftarrow s_{i_M} - \gamma/(2J)$     ▷ Projection
**end loop**

---

### 3.4 Parallel Implementation

The key realization which makes our barycenter algorithm truly scalable is that the variables $s, v^1, \ldots, v^J$ can be separated across different machines. In particular, the "sum" or "coupling" variable $s$ is maintained on a master thread which runs Algorithm 2, and each $v^j$ is maintained on a worker thread running Algorithm 3. Each projected gradient step requires first selecting distribution $j$. The algorithm then requires computing only $i_W = \text{argmin}_i \{c(x_j, y_i) - v_i^j\}$ and $i_M = \text{argmin}_i s_i$, and then updating $s$ and $v^j$ in only those coordinates. Hence only a small amount of information ($i_W$ and $i_M$) need pass between threads.

Note also that this algorithm can be adapted to the parallel shared-memory case, where $s$ is a variable shared between threads which make sparse updates to it. Here we will focus on the first master/worker scenario for simplicity.

Where are the bottlenecks? When there are $n$ points in the discrete approximation, each worker's task of computing $\text{argmin}_i \{c(x_j, y_i) - v_i^j\}$ requires $O(n)$ computations of $c(x, y)$. The master must iteratively find the minimum element $s_{i_M}$ in the vector $s$, then update $s_{i_M}$, and decrease element $s_{i_W}$. These can be implemented respectively as the "find min", "delete min" then "insert," and "decrease min" operations in a Fibonacci heap. All these operations together take amortized $O(\log n)$ time. Hence, it takes $O(n)$ time it for all $J$ workers to each produce one gradient sample in parallel, and only $O(J \log n)$ time for the master to process them all. Of course, communication is not free, but the messages are small and our approach should scale well for $J \ll n$.

This parallel algorithm is particularly well-suited to the Wasserstein posterior (WASP) [48] framework for merging Bayesian subset posteriors. In this setting, we split the dataset $X_1, \ldots, X_k$ into $J$ subsets $S_1, \ldots, S_J$ each with $k/J$ data points, distribute those subsets to $J$ different machines, then each machine runs Markov Chain Monte Carlo (MCMC) to sample from $p(\theta|S_i)$, and we aggregate these posteriors via their barycenter. The most expensive subroutine in the worker thread is actually sampling from the posterior, and everything else is cheap in comparison. In particular, the machines need not even share samples from their respective MCMC chains.

One subtlety is that selecting worker $j$ truly uniformly at random each iteration requires more synchronization, hence our gradient estimates are not actually independent as usual. Selecting worker threads as they are available will fail to yield a uniform distribution over $j$, as at the moment worker

$j$ finishes one gradient step, the probability that worker $j$ is the next available is much less than $1/J$: worker $j$ must resample and recompute $i_W$, whereas other threads would have a head start. If workers all took precisely the same amount of time, the ordering of worker threads would be determinstic, and guarantees for without-replacement sampling variants of stochastic gradient ascent would apply [44]. In practice, we have no issues with our approach.

## 4    Consistency

Prior methods for estimating the Wasserstein barycenter $\nu^*$ of continuous measures $\mu_j \in \mathcal{P}(\mathcal{X})$ involve first approximating each $\mu_j$ by a measure $\mu_{j,n}$ that has finite support on $n$ points, then computing the barycenter $\nu_n^*$ of $\{\mu_{j,n}\}$ as a surrogate for $\nu^*$. This approach is consistent, in that if $\mu_{j,n} \to \mu_j$ as $n \to \infty$, then also $\nu_n^* \to \nu^*$. This holds even if the barycenter is not unique, both in the Euclidean case [10, Theorem 3.1] as well as when $\mathcal{X}$ is a Riemannian manifold [28, Theorem 5.4]. However, it is not known how fast the approximation $\nu_n^*$ approaches the true barycenter $\nu^*$, or even how fast the barycentric distance $F[\nu_n^*]$ approaches $F[\nu_n]$.

In practice, not even the approximation $\nu_n^*$ is computed exactly: instead, support points are chosen and $\nu_n^*$ is constrained to have support on those points. There are various heuristic methods for choosing these support points, ranging from mesh grids of the support, to randomly sampling points from the convex hull of the supports of $\mu_j$, or even optimizing over the support point locations. Yet we are unaware of any rigorous guarantees on the quality of these approximations.

While our approach still involves approximating the barycenter $\nu^*$ by a measure $\nu_n^*$ with fixed support, we are able to provide bounds on the quality of this approximation as $n \to \infty$. Specifically, we bound the rate at which $F[\nu_n^*] \to F[\nu_n]$. The result is intuitive, and appeals to the notion of an $\epsilon$-cover of the support of the barycenter:

**Definition 4.1** (Covering Number). The $\epsilon$-*covering number* of a compact set $K \subset \mathcal{X}$, with respect to the metric $g$, is the minimum number $\mathcal{N}_\epsilon(K)$ of points $\{x_i\}_{i=1}^{\mathcal{N}_\epsilon(K)} \in K$ needed so that for each $y \in K$, there is some $x_i$ with $g(x_i, y) \leq \epsilon$. The set $\{x_i\}$ is called an $\epsilon$-covering.

**Definition 4.2** (Inverse Covering Radius). Fix $n \in \mathbb{Z}^+$. We define the *n-inverse covering radius* of compact $K \subset \mathcal{X}$ as the value $\epsilon_n(K) = \inf\{\epsilon > 0 : \mathcal{N}_\epsilon(K) \leq n\}$, when $n$ is large enough so the infimum exists.

---

**Algorithm 2** Master Thread

**Input:** index $j$, distribution $\mu$, atoms $\{y_i\}_{i=1,\dots,N}$, number $J$ of distributions, step size $\gamma$
**Output:** barycenter weights $w$
$c \leftarrow 0_n$
$s \leftarrow 0_n$
$i_M \leftarrow 1$
**loop**
    $i_W \leftarrow$ message from worker $j$
    Send $i_M$ to worker $j$
    $c_{i_M} \leftarrow c_{i_M} + 1$
    $s_{i_M} \leftarrow s_{i_M} + \gamma/(2J)$
    $s_{i_W} \leftarrow s_{i_W} - \gamma/2$
    $i_M \leftarrow \operatorname{argmin}_i s_i$
**end loop**
**return** $w \leftarrow c/(\sum_{i=1}^n c_i)$

---

**Algorithm 3** Worker Thread

**Input:** index $j$, distribution $\mu$, atoms $\{y_i\}_{i=1,\dots,N}$, number $J$ of distributions, step size $\gamma$
$v \leftarrow 0_n$
**loop**
    Draw $x \sim \mu$
    $i_W \leftarrow \operatorname{argmin}_i\{c(x, y_i) - v_i\}$
    Send $i_W$ to master
    $i_M \leftarrow$ message from master
    $v_{i_M} \leftarrow v_{i_M} + \gamma/(2J)$
    $v_{i_W} \leftarrow v_{i_W} - \gamma/2$
**end loop**

---

Suppose throughout this section that $K \subset \mathbb{R}^d$ is endowed with a Riemannian metric $g$, where $K$ has diameter $D$. In the specific case where $g$ is the usual Euclidean metric, there is an $\epsilon$-cover for $K$ with at most $C_1 \epsilon^{-d}$ points, where $C_1$ depends only on the diameter $D$ and dimension $d$ [43]. Reversing the inequality, $K$ has an $n$-inverse covering radius of at most $\epsilon \leq C_2 n^{-1/d}$ when $n$ takes the correct form.

We now present and then prove our main result:

**Theorem 4.1.** *Suppose the measures $\mu_j$ are supported on $K$, and suppose $\mu_1$ is absolutely continuous with respect to volume. Then the barycenter $\nu^*$ is unique. Moreover, for each empirical approximation size $n$, if we choose support points $\{y_i\}_{i=1,\dots,n}$ that constitute a $2\epsilon_n(K)$-cover of $K$, it follows that $F[\nu_n^*] - F[\nu^*] \leq O(\epsilon_n(K) + n^{-1/d})$, where $\nu_n^* = \sum_{i=1}^n w_i^* \delta_{y_i}$ for $w^*$ solving Problem* (8).

**Remark 4.1.** Absolute continuity is only needed to reason about approximating the barycenter with an $N$ point discrete distribution. If the input distributions are themselves discrete distributions,

so is the barycenter, and we can strengthen our result. For large enough $n$, we actually have $W_2(\nu_n^*, \nu^*) \leq 2\epsilon_n(K)$ and therefore $F[\nu_n^*] - F[\nu^*] \leq O(\epsilon_n(K))$.

**Corollary 4.1** (Convergence to $\nu^*$)**.** *Suppose the measures $\mu_j$ are supported on $K$, with $\mu_1$ absolutely continuous with respect to volume. Let $\nu^*$ be the unique minimizer of $F$. Then we can choose support points $\{y_i\}_{i=1,...,n}$ such that some subsequence of $\nu_n^* = \sum_{i=1}^n w_i^* \delta_{y_i}$ converges weakly to $\nu^*$.*

*Proof.* By Theorem 4.1, we can choose support points so that $F[\nu_n^*] \to F[\nu^*]$. By compactness, the sequence $\nu_n^*$ admits a convergent subsequence $\nu_{n_k}^* \to \nu$ for some measure $\nu$. Continuity of $F$ allows us to pass to the limit $\lim_{k \to \infty} F[\nu_{n_k}^*] = F[\lim_{k \to \infty} \nu_{n_k}^*]$. On the other hand, $\lim_{k \to \infty} F[\nu_{n_k}^*] = F[\nu^*]$, and $F$ is strictly convex [28], thus $\nu_{n_k}^* \to \nu^*$ weakly. □

Before proving Theorem 4.1, we need smoothness of the barycenter functional $F$ with respect to Wasserstein-2 distance:

**Lemma 4.1.** *Suppose we are given measures $\{\mu_j\}_{j=1}^J$, $\nu$, and $\{\nu_n\}_{n=1}^\infty$ supported on $K$, with $\nu_n \to \nu$. Then, $F[\nu_n] \to F[\nu]$, with $|F[\nu_n] - F[\nu]| \leq 2D \cdot W_2(\nu_n, \nu)$.*

*Proof of Theorem 4.1.* Uniqueness of $\nu^*$ follows from Theorem 2.4 of [28]. From Theorem 5.1 in [28] we know further that $\nu^*$ is absolutely continuous with respect to volume.

Let $N > 0$, and let $\nu_N$ be the discrete distribution on $N$ points, each with mass $1/N$, which minimizes $W_2(\nu_N, \nu^*)$. This distribution satisfies $W_2(\nu_N, \nu^*) \leq CN^{-1/d}$ [30], where $C$ depends on $K$, the dimension $d$, and the metric. With our "budget" of $n$ support points, we can construct a $2\epsilon_n(K)$-cover as long as $n$ is sufficiently large. Then define a distribution $\nu_{n,N}$ with support on the $2\epsilon_n(K)$-cover as follows: for each $x$ in the support of $\nu_N$, map $x$ to the closest point $x'$ in the cover, and add mass $1/N$ to $x'$. Note that this defines not only the distribution $\nu_{n,N}$, but also a transport plan between $\nu_N$ and $\nu_{n,N}$. This map moves $N$ points of mass $1/N$ each a distance at most $2\epsilon_n(K)$, so we may bound $W_2(\nu_{n,N}, \nu_N) \leq \sqrt{N \cdot 1/N \cdot (2\epsilon_n(K))^2} = 2\epsilon_n(K)$. Combining these two bounds, we see that

$$W_2(\nu_{n,N}, \nu^*) \leq W_2(\nu_{n,N}, \nu_N) + W_2(\nu_N, \nu^*) \tag{13}$$

$$\leq 2\epsilon_n(K) + CN^{-1/d}. \tag{14}$$

For each $n$, we choose to set $N = n$, which yields $W_2(\nu_{n,n}, \nu^*) \leq 2\epsilon_n(K) + Cn^{-1/d}$. Applying Lemma 4.1, and recalling that $\nu^*$ is the minimizer of $J$, we have

$$F[\nu_{n,n}] - F[\nu^*] \leq 2D \cdot (2\epsilon_n(K) + Cn^{-1/d}) = O(\epsilon_n(K) + n^{-1/d}). \tag{15}$$

However, we must have $F[\nu_n^*] \leq F[\nu_{n,n}]$, because both are measures on the same $n$ point $2\epsilon_n(K)$-cover, but $\nu_n^*$ has weights chosen to minimize $J$. Thus we must also have

$$F[\nu_n^*] - F[\nu^*] \leq F[\nu_{n,n}] - F[\nu^*] \leq O(\epsilon_n(K) + n^{-1/d}). \qquad □$$

The high-level view of the above result is that choosing support points $y_i$ to form an $\epsilon$-cover with respect to the metric $g$, and then optimizing over their weights $w_i$ via our stochastic algorithm, will give us a consistent picture of the behavior of the true barycenter. Also note that the proof above requires an $\epsilon$-cover only of the support of $v^*$, not all of $K$. In particular, an $\epsilon$-cover of the convex hull of the supports of $\mu_j$ is sufficient, as this must contain the barycenter. Other heuristic techniques to efficiently focus a limited budget of $n$ points only on the support of $\nu^*$ are advantageous and justified.

While Theorem 4.1 is a good start, ideally we would also be able to provide a bound on $W_2(\nu_n^*, \nu^*)$. This would follow readily from sharpness of the functional $F[\nu]$, or even the discrete version $F(w)$, but it is not immediately clear how to achieve such a result.

## 5   Experiments

We demonstrate the applicability of our method on two experiments, one synthetic and one performing a real inference task. Together, these showcase the positive traits of our algorithm: speed, parallelization, robustness to non-stationarity, applicability to non-Euclidean domains, and immediate performance benefit to Bayesian inference. We implemented our algorithm in C++ using MPI, and our code is posted at github.com/mstaib/stochastic-barycenter-code. Full experiment details are given in Appendix D.

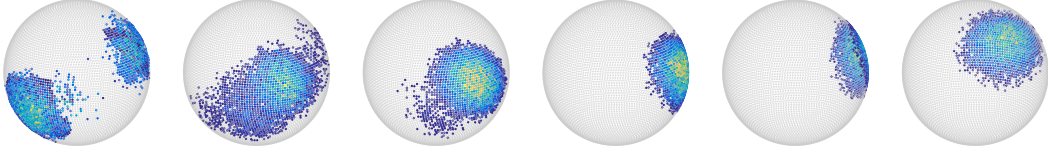

Figure 1: The Wasserstein barycenter of four von Mises-Fisher distributions on the unit sphere $S^2$. From left to right, the figures show the initial distributions merging into the Wasserstein barycenter. As the input distributions are moved along parallel paths on the sphere, the barycenter accurately tracks the new locations as shown in the final three figures.

## 5.1 Von Mises-Fisher Distributions with Drift

We demonstrate computation and tracking of the barycenter of four drifting von Mises-Fisher distributions on the unit sphere $S^2$. Note that $W_2$ and the barycentric cost are now defined with respect to geodesic distance on $S^2$.

The distributions are randomly centered, and we move the center of each distribution $3 \times 10^{-5}$ radians (in the same direction for all distributions) each time a sample is drawn. A snapshot of the results is shown in Figure 1. Our algorithm is clearly able to track the barycenter as the distributions move.

## 5.2 Large Scale Bayesian Inference

We run logistic regression on the UCI skin segmentation dataset [8]. The 245057 datapoints are colors represented in $\mathbb{R}^3$, each with a binary label determing whether that color is a skin color. We split consecutive blocks of the dataset into 127 subsets, and due to locality in the dataset, the data in each subsets is *not* identically distributed. Each subset is assigned one thread of an InfiniBand cluster on which we simultaneously sample from the subset posterior via MCMC and optimize the barycenter estimate. This is in contrast to [47], where the barycenter can be computed via a linear program (LP) only after all samplers are run.

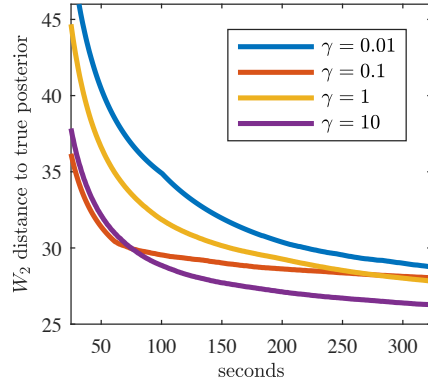

Figure 2: Convergence of our algorithm with $n \approx 10^4$ for different stepsizes. In each case we recover a better approximation than what was possible with the LP for any $n$, in as little as $\approx 30$ seconds.

Since the full dataset is tractable, we can compare the two methods via $W_2$ distance to the posterior of the full dataset, which we can estimate via the large-scale optimal transport algorithm in [22] or by LP depending on the support size. For each method, we fix $n$ barycenter support points on a mesh determined by samples from the subset posteriors. After 317 seconds, or about 10000 iterations per subset posterior, our algorithm has produced a barycenter on $n \approx 10^4$ support points with $W_2$ distance about 26 from the full posterior. Similarly competitive results hold even for $n \approx 10^5$ or $10^6$, though tuning the stepsize becomes more challenging. Even in the $10^6$ case, no individual 16 thread node used more than 2GB of memory. For $n \approx 10^4$, over a wide range of stepsizes we can in seconds approximate the full posterior better than is possible with the LP as seen in Figure 2 by terminating early.

In comparison, in Table 1 we attempt to compute the barycenter LP as in [47] via Mosek [4], for varying values of $n$. Even $n = 480$ is not possible on a system with 16GB of memory, and feasible values of $n$ result in meshes too sparse to accurately and reliably approximate the barycenter. Specifically, there are several cases where $n$ increases but the approximation quality actually decreases: the subset posteriors are spread far apart, and the barycenter is so small relative to the required bounding box that likely only one grid point is close to it, and how close this grid point is depends on the specific mesh. To avoid this behavior, one must either use a dense grid (our approach), or invent a better method for choosing support points that will still cover the barycenter. In terms of compute time, entropy regularized methods may have faired better than the LP for finer meshes but would still

Table 1: Number of support points $n$ versus computation time and $W_2$ distance to the true posterior. Compared to prior work, our algorithm handles much finer meshes, producing much better estimates.

| | Linear program from [47] | | | | | | | | This paper |
|---|---|---|---|---|---|---|---|---|---|
| $n$ | 24 | 40 | 60 | 84 | 189 | 320 | 396 | 480 | $10^4$ |
| time (s) | 0.5 | 0.97 | 2.9 | 6.1 | 34 | 163 | 176 | out of memory | 317 |
| $W_2$ | 41.1 | 59.3 | 50.0 | 34.3 | 44.3 | 53.7 | 45 | out of memory | 26.3 |

not give the same result as our method. Note also that the LP timings include only optimization time, whereas in 317 seconds our algorithm produces samples *and* optimizes.

## 6 Conclusion and Future Directions

We have proposed an original algorithm for computing the Wasserstein barycenter of arbitrary measures given a stream of samples. Our algorithm is communication-efficient, highly parallel, easy to implement, and enjoys consistency results that, to the best of our knowledge, are new. Our method has immediate impact on large-scale Bayesian inference and sensor fusion tasks: for Bayesian inference in particular, we obtain far finer estimates of the Wasserstein-averaged subset posterior (WASP) [47] than was possible before, enabling faster and more accurate inference.

There are many directions for future work: we have barely scratched the surface in terms of new applications of large-scale Wasserstein barycenters, and there are still many possible algorithmic improvements. One implication of Theorem 3.1 is that a faster algorithm for solving the concave problem (11) immediately yields faster convergence to the barycenter. Incorporating variance reduction [18, 27] is a promising direction, provided we maintain communication-efficiency. Recasting problem (11) as distributed consensus optimization [35, 11] would further help scale up the barycenter computation to huge numbers of input measures.

**Acknowledgements**  We thank the anonymous reviewers for their helpful suggestions. We also thank MIT Supercloud and the Lincoln Laboratory Supercomputing Center for providing computational resources. M. Staib acknowledges Government support under and awarded by DoD, Air Force Office of Scientific Research, National Defense Science and Engineering Graduate (NDSEG) Fellowship, 32 CFR 168a. J. Solomon acknowledges funding from the MIT Research Support Committee ("Structured Optimization for Geometric Problems"), as well as Army Research Office grant W911NF-12-R-0011 ("Smooth Modeling of Flows on Graphs"). This research was supported by NSF CAREER award 1553284 and The Defense Advanced Research Projects Agency (grant number N66001-17-1-4039). The views, opinions, and/or findings contained in this article are those of the author and should not be interpreted as representing the official views or policies, either expressed or implied, of the Defense Advanced Research Projects Agency or the Department of Defense.

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
