[Supplementary Material]

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

# A   Sparse Projections

Our algorithms for solving the barycenter problem in the parallel setting relied on the ability to efficiently project the matrix $A = (s, v^1, \ldots, v^J)$ back onto the constraint set $s = \sum_{j=1}^J v^j$. For the sake of completion, we include a proof that our sparse updates actually result in projection with respect to the $\ell_1$ norm.

At any given iteration of gradient ascent, we start with some iterate $A = (s, v^1, \ldots, v^J)$ which does satisfy the constraint. Suppose we selected distribution $j$. The gradient estimate is a sparse $n \times (J+1)$ matrix $M$ which has $M_{u1} = 1/J$ and $M_{vj} = -1$, for some indices $u$ and $v$, with column 1 corresponding to $s$ and column $j$ corresponding to $v^j$. After the gradient step with stepsize $\gamma$, we have $A + \gamma M$. Now, our constraint can be written in matrix form as $Az = 0$, where

$$z = \begin{bmatrix} -1 \\ 1 \\ \vdots \\ 1 \end{bmatrix}, \tag{16}$$

and so the problem of projecting $A + \gamma M$ onto this constraint set can be written as

$$\begin{aligned} \min_B \quad & \|A + \gamma M - B\|_1 \\ \text{s.t.} \quad & Bz = 0. \end{aligned} \tag{17}$$

Equivalently, we want to find the matrix $C$ solving

$$\begin{aligned} \min_C \quad & \|C\|_1 \\ \text{s.t.} \quad & (A + \gamma M + C)z = 0. \end{aligned} \tag{18}$$

Note that

$$(A + \gamma M + C)z = 0 \Leftrightarrow Cz = -\gamma Mz = \gamma\left(\frac{1}{J}e_u + e_v\right). \tag{19}$$

Consider the sparse matrix $C$ given by $C_{u1} = -\gamma/(2J)$, $C_{uj} = \gamma/(2J)$, $C_{v1} = \gamma/2$, and $C_{vj} = -\gamma/2$. Define a sparse vector $\lambda \in \mathbb{R}^n$ by $\lambda_u = \lambda_v = -1$. We wish to show that the primal dual pair $(C, \lambda)$ solves problem (18). We can do this directly by looking at the Karush–Kuhn–Tucker conditions. It is easy to check that $C$ is primal feasible, so it remains only to show that

$$0 \in \partial_C(\|C\|_1 + \lambda^T Cz) \Leftrightarrow -z\lambda^T \in \partial_C(\|C\|_1). \tag{20}$$

The subgradients of the $\ell_1$ norm at $C$ are matrices $G$ which satisfy $\|G\|_\infty \leq 1$ and $\langle G, C\rangle = \|C\|_1$. It is easy to check that $\|z\lambda^T\|_\infty = 1$. Finally,

$$\langle -z\lambda^T, C\rangle = -\lambda^T Cz = -\gamma\lambda^T\left(\frac{1}{J}e_u + e_v\right) \tag{21}$$

$$= \gamma \cdot \left(\frac{1}{J} + 1\right) \tag{22}$$

$$= \|C\|_1. \tag{23}$$

Hence after the gradient step we can project onto the feasible set with respect to $\ell_1$, simply by adding the sparse matrix $C$.

# B   Stochastic Gradient Bound

We first need a lemma which gives a regret bound for online gradient ascent:

**Lemma B.1** (Adapted from [23, Theorem 3.1]). *Run online gradient ascent on concave functions $f_t$ with subgradients $g_t \in \partial f_t(x_t)$. Assume $\|x_t - x^*\| \leq R$ for some optimizer $x^*$ of $\sum_{t=1}^T f_t$, and assume $\mathbb{E}[\|g_t\|] \leq G$. Using stepsize $\gamma = \frac{R}{G\sqrt{T}}$, the expected regret after $T$ iterations is bounded by $2RG\sqrt{T}$.*

*Proof of Theorem 3.1.* This is adapted from [20, 57].

Define $f(s, v, w) = \langle s, w\rangle + \frac{1}{J}\sum_{j=1}^J \mathbb{E}_{X_j \sim \mu_j}[h(X_j, v^j)]$ as in (11). For simplicity, concatenate $s$ and $v$ into a vector $u$, with $f(u, w) = f(s, v, w)$. Write $w^*(u) = \operatorname{argmin}_{w \in \Delta_n}\langle s, w\rangle$ and note that our objective in Equation (11) is $f(u) := f(u, w^*(u))$.

Recall the online optimization setup: at time step $t$ we play $u_t$, then receive $f_t$ and reward $f_t(u_t)$, then update $u_t$ and repeat. Note that if $f_t$ is given by $f(u_t, w^*(u_t))$, then online gradient ascent on $f_t$ is effectively subgradient ascent on $f$. Suppose we play online subgradient ascent and achieve average expected regret $\varepsilon(T)$ after $T$

timesteps, where the expectation is with respect to the gradient estimates in the learning algorithm. Then by the definition of expected regret,

$$\varepsilon(T) \geq \mathbb{E}\left[\sup_u \frac{1}{T}\sum_{t=1}^{T} f_t(u) - \frac{1}{T}\sum_{t=1}^{T} f_t(u_t)\right] = \mathbb{E}\left[\sup_v \frac{1}{T}\sum_{t=1}^{T} f(u,w_t) - \frac{1}{T}\sum_{t=1}^{T} f(u_t,w_t)\right]. \quad (24)$$

where we write $w_t = w^*(u_t)$. Simultaneously, we have

$$\frac{1}{T}\sum_{t=1}^{T} f(u_t,w_t) - \inf_w \frac{1}{T}\sum_{t=1}^{T} f(u_t,w) \leq \frac{1}{T}\sum_{t=1}^{T} f(u_t,w_t) - \frac{1}{T}\sum_{t=1}^{T} f(u_t,w_t) = 0 \quad (25)$$

because $w_t$ are each chosen optimally. Summing, we have

$$\mathbb{E}\left[\sup_u \frac{1}{T}\sum_{t=1}^{T} f(u,w_t) - \inf_w \frac{1}{T}\sum_{t=1}^{T} f(u_t,w)\right] \leq \varepsilon(T). \quad (26)$$

Now we merely need combine this with the standard bound:

$$\inf_w \frac{1}{T}\sum_{t=1}^{T} f(u_t,w) \leq \inf_w f(\overline{u}_T,w) \leq \sup_v \inf_w f(u,w) \quad (27)$$

$$\leq \inf_w \sup_u f(u,w) \leq \sup_u f(u,\overline{w}_T) \leq \sup_u \frac{1}{T}\sum_{t=1}^{T} f(u,w_t). \quad (28)$$

The extreme bounds on either side of this chain of inequalities are within $\varepsilon(T)$, hence we also have

$$\mathbb{E}\left[\sup_u f(u,\overline{w}_T) - \inf_w \sup_u f(u,w)\right] \leq \varepsilon(T). \quad (29)$$

By definition of $f$, the left hand side is precisely $\mathbb{E}[F(\overline{w}_T) - F(w^*)]$. Now, noting that our gradient estimates $g$ are always sparse (we always have two elements of magnitude 1, so $\|g\|_1 = 2$), we simply replace $\varepsilon(T)$ with the particular regret bound of Lemma B.1 for online gradient ascent. $\square$

## C  Smoothness of barycenter functional

*Proof of Lemma 4.1.* For any two measures $\eta, \eta'$ supported on $K$, we can bound $W_2(\eta,\eta') \leq D$: the worst-case $\eta, \eta'$ are point masses distance $D$ apart, so that the transport plan sends all the mass a distance of $D$.

It follows that $|W_2(\mu,\nu_n) + W_2(\mu,\nu)| \leq 2D$ and therefore

$$|W_2^2(\mu,\nu_n) - W_2^2(\mu,\nu)| \leq 2D \cdot |W_2(\mu,\nu_n) - W_2(\mu,\nu)| \quad (30)$$

$$\leq 2D \cdot W_2(\nu_n,\nu) \quad (31)$$

by the triangle inequality. Summing over all $\mu = \mu_j$, we find that

$$|F[\nu_n] - F[\nu]| \leq \frac{1}{J}\sum_{j=1}^{J} |W_2^2(\mu_j,\nu_n) - W_2^2(\mu_j,\nu)| \quad (32)$$

$$\leq \frac{1}{J}\sum_{j=1}^{J} 2D \cdot W_2(\nu_n,\nu) = 2D \cdot W_2(\nu_n,\nu), \quad (33)$$

completing the proof. $\square$

## D  Experiment details

### D.1  Von Mises-Fisher Distributions with Drift

The distributions are randomly centered with concentration parameter $\kappa = 30$. To verify that the barycenter accurately tracks when the input distributions are non-stationary, we move the center of each distribution $3 \times 10^{-5}$ radians (in the same direction for all distributions) each time a sample is drawn. A snapshot of the results is shown in Figure 1.

We use a sliding window of $T = 10^5$ timesteps with step size $\gamma = 1$ and on $N = 10^4$ evenly-distributed support points. Each thread is run for $5 \times 10^5$ iterations on a separate thread of an 8 core workstation. The total time is roughly 80 seconds, during which our algorithm has processed a total of $2 \times 10^6$ samples. Clearly our algorithm is efficient and is able to perform the specified task.

### D.2 Large Scale Bayesian Inference

**Subset assignment.** The skin segmentation dataset is given with positive samples grouped all together, then negative samples grouped together. To ensure even representation of positive and negative samples across all subsets, while simulating the non-i.i.d data setting, each subset is composed of a consecutive block of positive samples and one of negative samples.

**MCMC chains.** We used a simple Metropolis-Hastings sampler with Gaussian proposal distribution $\mathcal{N}(0, \sigma^2 I)$, for $\sigma = 0.05$. We used a very conservative $10^5$ burn-in iterations, and afterwards took every fifth sample.

**Mesh selection.** During the burn-in phase, we compute a minimum axis-aligned bounding box containing all samples from all MCMC chains. Then, for a desired mesh size of $n$, we choose a granularity $\delta$ so that cutting each axis into evenly-spaced values differing by $\delta$ results in approximately $n$ points total. For Table 1 in particular, the bounding box was selected as roughly

$$[-21.4, 114.4] \times [-121.8, 42.9] \times [-50.8, 6.1]$$

for the LP code, and

$$[-21.4, 114.2] \times [-121.5, 42.8] \times [-50.7, 6.1]$$

for an arbitrary $n \approx 10^4$ instance of our stochastic algorithm. That these match so well supports the consistency of our implementation. The griddings assigned for the LPs are given in Table 2.

Table 2: Grid sizes chosen for LP experiments.

| $n$ | grid dimensions |
|-----|-----------------|
| 24  | $3 \times 4 \times 2$ |
| 40  | $4 \times 5 \times 2$ |
| 60  | $5 \times 6 \times 2$ |
| 84  | $6 \times 7 \times 2$ |
| 189 | $7 \times 9 \times 3$ |
| 320 | $8 \times 10 \times 4$ |
| 396 | $9 \times 11 \times 4$ |
| 480 | $10 \times 12 \times 4$ |

**Optimization.** We experimented with different stepsizes in $\{0.01, 0.1, 1, 10, 100\}$ for $n$ in $\{10^3, 10^4, 10^5, 10^6\}$. As expected, more aggressive step sizes are needed as $n$ grows, but competitive barycenter estimates were possible for all $n$ given sufficient iterations. The 317 seconds value corresponds to $127 \times 10^4$ iterations total, or 10000 per sampler. The barycenter estimate $\overline{w}_T = \frac{1}{T} \sum_{t=1}^{T} e_{i_t}$ was maintained over all $127 \times 10^4$ iterations. We found that it is sometimes helpful to use a sliding window, to quicker move away from initial bad barycenter estimates.

**Error metric.** We stored $10^4$ samples from the true posterior, and computed the $W_2$ distance between these samples and each candidate barycenter.