[Reviews · NeurIPS 2017]

Reviewer 1



The paper ‘Parallel streaming Wasserstein Barycenters’ presents a method for scaling the computation of Wasserstein barycenters in a distributed and principled way. Through a semi-discrete formulation of the problem allowing to handle continuous and nonstationary distributions. Bounds are provided on the recovery of the true barycenter wrt. discretization and its effectivity is demonstrated on a toy example and a large scale Bayesian inference problem. The paper is very well-written and contributes strongly to the use of Wasserstein distance in large scale settings. The methodological and theoretical contributions are strong. The use of the method in a WASP setting [47] is particularly appealing and promising. For those reasons I will recommend a strong accept. I have some small questions/remarks that coud be addressed in a revised version provided that the paper is accepted: - it is not clear if problem (10) and (11) are strictly equivalent. If yes, could you discuss why ? Can you comment on the ‘sparsity of gradients’ as discussed above ? Why is it an issue ? - as in [22], computation of i_W can be done in closed-form if the W distance is regularized by entropy. What is the impact of the overall complexity of the parallel implementation ? Did the authors experiment this ? - what is an InfiniBand cluster ? - in table 1, with the LP solver from [47], why is the W_2 distance is not decreasing monodically wrt. n ? After rebuttal. I wish to thank the authors for their explanations and clarifications

Reviewer 2



Title: Parallel Streaming Wasserstein Barycenters Comments: - This paper presents a new method for performing low-communication parallel inference via computing the Wasserstein barycenter of a set of distributions. Unlike previous work, this method aims to reduce certain approximations incurred by discretization. Theoretically, this paper gives results involving the rate of the convergence of the barycenter distance. Empirically, this paper shows results on a synthetic task involving a Von Mises distribution and on a logistic regression task. - I feel that the clarity of writing in this paper is not great. It would be better to clearly (and near the beginning of the paper) give an intuition behind the methodology improvements that this paper aims to provide, relative to previous work on computing the Wasserstein barycenter. This paper quickly dives into the algorithm and theory details (“Background”, “Mathematical preliminaries”, “Deriving the optimization problem”), without giving a clear treatment of previous work, and how this methods of this paper differ from this work. It is therefore is hard to see where the material developed in previous work ends and the new methodology of this paper begins. A simple description (early on) of this method, how it differs from existing methods, and why it solves the problem inherent in these existing methods, would greatly increase the clarity of this paper. - Furthermore, it would be nice to include more motivation behind the main theoretical results that are proved in this paper. It is hard to get a grasp on the usefulness and contribution of these results without some discussion or reasoning on why one would like to prove these results (e.g. the benefits of proving this theory). - Finally, I do not think that the empirical results in this paper are particularly thorough. The results in Table 1 are straightforward, but these seem to be the only empirical argument of the paper, and they are quite minimal.

Reviewer 3



This paper applies optimal transport algorithm to the barycenter communication problem. It approximates continuous distributions with discrete distributions with finite support and use stochastic projected gradient decent to solve the optimization problem. It shows the consistency of this algorithm and parallelizability of this algorithm because the variables can be decoupled. The authors show the speedup and improved memory consumption of this new algorithm. Also it works with nonstationary data. Overall a nice paper. It could be interesting to see how the accuracy of this new algorithm compared to LP. How much do we lose in adopting this new algorithm? Or do we ever lose anything? ---------------------- After author rebuttal: Many thanks for the rebuttal from the authors. They do clarify. My scores remain the same though. Thanks!